# NEGATIVE EIGENVALUES OF THE HESSIAN IN DEEP NEURAL NETWORKS

**Guillaume Alain** *
MILA
University of Montreal
guillaume.alain.
  umontreal@gmail.com

**Nicolas Le Roux**
Google Brain
Montreal
nicolas@le-roux.name

**Pierre-Antoine Manzagol**
Google Brain
Montreal
manzagop@google.com

## ABSTRACT

We study the loss function of a deep neural network through the eigendecomposition of its Hessian matrix. We focus on negative eigenvalues, how important they are, and how to best deal with them. The goal is to develop an optimization method specifically tailored for deep neural networks.

## 1 INTRODUCTION

The current mode of operation in the field of Deep Learning is that we accept the fact that saddle points are everywhere (Choromanska et al., 2015) and that many local minima are of such high quality that we do not need to worry about not having the global minimum. Practitioners sweep a large collection of hyperparameter configurations, they use early stopping to prevent overfitting, and they train their models with optimization methods such as RMSProp (Tieleman & Hinton, 2012) and ADAM (Kingma & Ba, 2015).

Most optimization methods used in deep learning today were developed with the convex setting in mind. We currently do not have an efficient way to specifically manage the negative eigenvalues of the Hessian matrix (which contains the second order derivatives and describes the curvature of the loss). We want to develop specific methods adapted to our particular kind of non-convex problems. Such methods will handle regions of negative curvature in a particular way, because this phenomenon is not present in convex optimization.

We present here experimental results that

- help us better understand what is happening in the directions of negative curvature,
- suggest that we should be using a much larger step size in those directions.

## 2 EXPERIMENTS

### 2.1 METHODOLOGY

Since we are working purely in an optimization context, we are not interested in the generalization error. We want to focus on the challenges of minimizing a loss that features saddle points and local minima.

The size of the Hessian matrix scales proportionally to the square of the number of parameters, so there is no way to compute and store the entire Hessian. We can still extract certain properties of the Hessian despite this, but we find ourselves limited to smaller models and datasets.

We are going to use the architecture of the classic LeNet (LeCun et al., 1989) convolution neural network, but with ReLU as activation function. It has two convolutional layers, two fully connected layers, and a softmax on the last layer, for a total number of approximately $d = 3.3 \times 10^6$ parameter coefficients. We performed experiments with MNIST (LeCun, 1998).

---

*This work was done during an internship with the Google Brain team in Montreal.

We have to keep in mind that there is no guarantee that phenomena observed in this setup will also be found in a much larger convolutional neural network such as Inception (Szegedy et al., 2015), or one with a different design such as ResNet (He et al., 2016).

While we are training our model using the typical minibatch gradient descent with RMSProp (batch size 32), it makes sense for our analysis to study the loss $\mathcal{L}(\theta)$ averaged over the full training set instead of minibatches. The same applies for the gradient $g(\theta) \in \mathbb{R}^d$ and the Hessian matrix $H(\theta) \in \mathbb{R}^{d \times d}$. We made the decision to concatenate all the parameters from all layers into a single vector in $\mathbb{R}^d$. Though results on the Hessian of individual layers were not included in this study, we believe they would also be of interest for a better understanding of deep neural networks.

Note that all the eigenvalues are real-valued because of the symmetry of the Hessian matrix, so they can be ordered as $\lambda_1 \geq \lambda_2 \geq \ldots \geq \lambda_d$. See Appendix section A for details on how we can compute the $k$ largest or smallest eigenpairs $(\lambda_i, v_i)$.

## 2.2 NEGATIVE CURVATURE IS ONLY LOCAL

At any training step $t$, we can select an eigenpair $(\lambda_i, v_i)$ and measure the loss function when we project the gradient $g(\theta_t)$ in the direction $v_i$. With a step size of $\alpha \in \mathbb{R}$, we look at

$$\mathcal{L}(\theta_t - \alpha \left[ g(\theta)^T v_i \right] v_i). \tag{1}$$

This becomes particularly interesting when $\lambda_i$ is negative and when we make the mild assumption that $v_i$ in not perfectly orthogonal to the gradient (i.e. $g(\theta)^T v_i \neq 0$).

Since we observed a common behaviour all along the optimization, we show here the results for an arbitrary iteration ($t = 50$). We use $\alpha \in [-0.1, 0.1]$ in Figure 1 and $\alpha \in [-1, 1]$ in Figure 2. We compare the exact empirical loss (orange curve) alongside the quadratic approximation (green/blue curve) of the same function given by the negative eigenvalue $\lambda_i$.

For small values of $\alpha$, the actual loss matches the curvature sufficiently well, but for larger values of $\alpha$ the two are qualitatively different. Because the loss is bounded below, it would be impossible for the loss to go down to $-\infty$. When using a regularizer such as an L2-norm penalty, the loss grows to $\infty$ when $\|\theta\| \to \infty$.

Note that, if we were to optimize for long enough, we would get into the neighborhood of a local minimum and we would not observe any negative eigenvalues anymore. In that later regime, there is nothing to gain from having an optimizer designed to deal with negative eigenvalues. However, there are no theoretical results clarifying when that regime starts. In practice, when early stopping is used as an approach to avoid overfitting, is it also unclear in what regime we stop training.

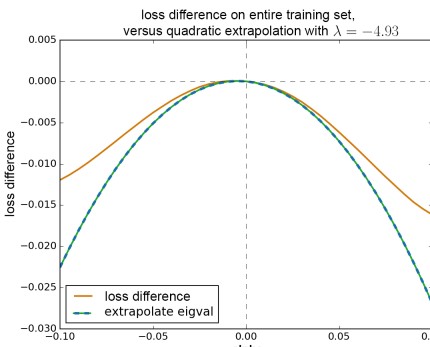

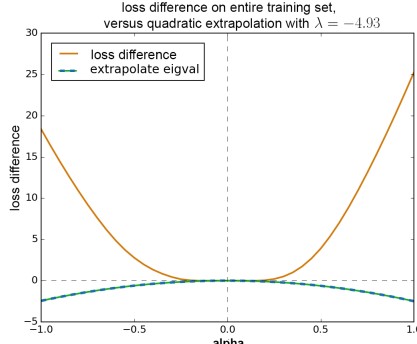

Figure 1: Looking at the total loss when moving by $\alpha$ in the direction of most negative curvature. Evaluated at training step $t = 50$. Zoomed in.

Figure 2: Same direction of negative curvature as Figure 1, but zoomed out.

### 2.3 MINIMIZING LOSS IN DIRECTIONS OF NEGATIVE CURVATURE

What is the connection between $\lambda_i$ and the optimal step size to be taken in the direction of $v_i$?

We go back to the question of finding the optimal $\alpha$ to minimize the line search problem in Equation (1). It is simple enough (albeit costly) to run through the whole training set and evaluate the loss at multiple values of $\alpha$, spanning a few orders of magnitude. For all the eigenpairs $(\lambda_i, v_i)$ that we have access to, we can look at

- what is the best loss decrease that we can obtain by moving along $v_i$? (see Figure 3)
- what is the optimal step size $\alpha^*$ to achieve it? (see Figure 4)

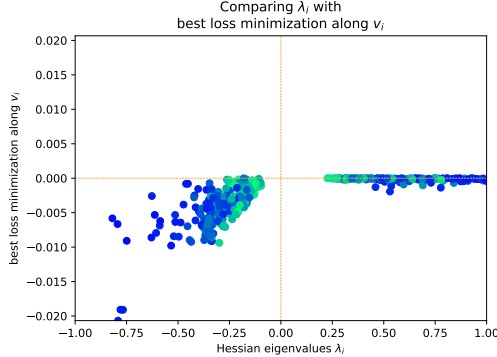 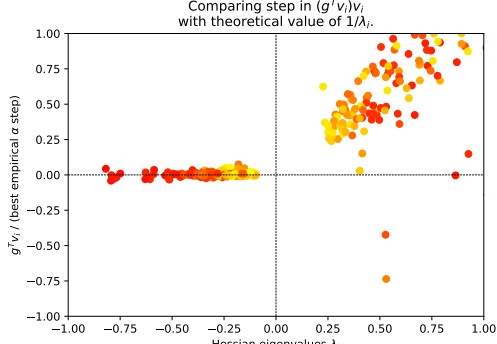

Figure 3: Best loss decrease possible ($y$-axis) when following eigenvector associated with $\lambda$ ($x$-axis). Lower is better. Directions of negative curvature (left side) were empirically observed to bring larger improvements in the loss than directions of positive curvature (right side). Earlier time steps $t$ are shown blue, and later are shown in green. In terms of Equation (1), this plot shows the relation between $\lambda_i$ and $\mathcal{L}(\alpha^*)$.

Figure 4: Reporting the actual optimal step sizes found empirically. In terms of the variables involved in Equation (1), this plot shows the relation between $\lambda_i$ ($x$-axis) and $1/\alpha^*$ ($y$-axis). On the right side of the plot, we can report that in direction of positive curvature we have that $1/\alpha^* \approx \lambda_i$. On the left side of the plot, the small values reported mean that the optimal step sizes were quite large. Earlier time steps $t$ are shown red, and later are shown in yellow.

Figures 3 and 4 suggest that important gains are to be made in directions of negative curvature, and that in directions of negative curvature the optimal step sizes are of a greater order of magnitude than in directions of positive curvature. Refer to Appendix section C for a longer discussion about optimal step sizes. Note that in Figures 3 and 4 we are showing a certain range where we find eigenvalues $\lambda \in [-1, 1]$. This is the most informative plot for us, but are not showing everything here. Keep in mind also that we are using numerical methods that report eigenvalues with the largest magnitude $|\lambda|$, so those figures are missing more than 99.99% of the eigenvalues with very small magnitude. This is why those figures do not have any points shown around the origin.

## 3 FUTURE WORK AND CONCLUSION

The current results need to be validated in more settings of architectures and optimizers.

Considerable work was required for us to extract negative eigenvalues for every checkpoint of training. This is not a pratical thing to do during training. In Appendix E we propose a new method that maintains an estimate of the most negative eigenvector and uses it to update the parameters. We have not yet tried this method in practice.

The main contribution of our work is that we have observed and studied an example where the directions of negative curvature are not being exploited properly by the popular convex optimizer. We have seen how great gains could be made in those directions. This reinforces the belief that there are opportunities to develop new optimization techniques that capitalize on the specific case of neural networks.

ACKNOWLEDGMENTS

We thank Bart van Merriënboer for fruitful discussions about optimization and the problem of saddle points. We thank Ying Xiao for initial discussions about his preliminary work on studying eigenvalues, and for providing his code to quickly get the "Jacobian vector product" trick working.

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

## A  JACOBIAN VECTOR PRODUCT

With $d = 3.3 \times 10^6$, the storage required to store the symmetric Hessian matrix with float32 coefficients is approximately 20 terabytes, which makes it close to impossible to store in RAM. The task of computing all the $d$ eigenvalues is absolutely out of reach, but by using the "Jacobian Vector Product" trick (Townsend, 2017), along with Scipy (Jones et al., 2014; Lehoucq et al., 1998), we can compute the $k$ largest or smallest eigenpairs $(\lambda_i, v_i)$.

The Scipy library function `scipy.sparse.linalg.eigsh` is able to accept either a symmetric matrix, or a function that computes the product $v \mapsto H(\theta)v$. We define a Python function that makes many internal calls to Tensorflow to iterate over the whole training set (or a fixed subset thereof). We aggregate the results and return them. This enables a Scipy library function to make calls to Tensorflow without being aware of it.

Following again the notation section 2.1, we order the eigenvalues as $\lambda_1 \geq \lambda_2 \geq \ldots \geq \lambda_d$. They are all real-valued because the Hessian matrix is symmetric and contains only real coefficients.

We are mainly interested in the eigenvalues closest to $\pm\infty$, so we define the following notation to refer to the $k$ most extreme eigenpairs on each side.

$$\begin{aligned}
\text{LA}(k) &= \{(\lambda_1, v_1), \ldots, (\lambda_k, v_k)\} \\
\text{SA}(k) &= \{(\lambda_{d-k+1}, v_{d-k+1}), \ldots, (\lambda_d, v_d)\}.
\end{aligned}$$

Note that the costs of computing those sets depends a lot of the magnitude of the eigenvalues. In practice we observed that the LA eigenvalues have a much larger magnitude than the SA (see Appendix B). This leads to the task of computing LA(20) being much cheaper than LA(3), despite the fact that it involves more eigenvalues.

For reasons of computational costs, we resorted to using a fixed subset of the training set when we performed the eigendecompositions (done after training).

## B    PROGRESSION OF EIGENVALUES DURING TRAINING

In Figure 5 and Figure 6 we show the evolution of the eigenvalues on models training on MNIST and CIFAR10. The largest eigenvalues LA(20) are shown in blue/green, while the smallest eigenvalues SA(3) are shown in red. We use log-scale on the vertical axis, so this means that the smallest eigenvalues, which are negative, are plotted as $\log|\lambda|$. We also report the total loss as a dotted black curve, also in log-scale, with their scale shown in the right side of the figures.

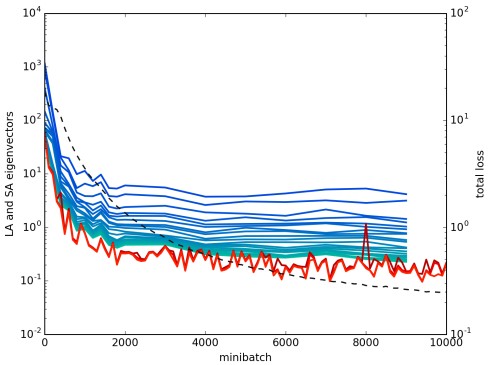
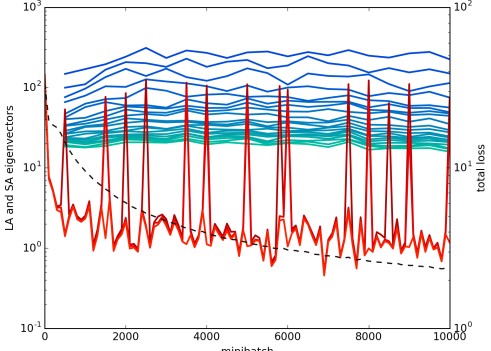

Figure 5:  $\log|\lambda|$ for eigenvalues of Hessian on MNIST, computed separately at many moments of training.  Largest in blue/green, smallest in red.

Figure 6:  Same as left figure, but with CIFAR10.  We are not sure how to interpret the spikes.

The first surprising observation that we made was that the largest and smallest eigenvalues are not affected a lot during training.  They stabilize very quickly (2000 minibatches of size 32 is only slightly more than a full epoch) while the loss is still in the process in being minimized.  Note that this does not necessarily mean that the leading eigenvector $v_1$ stays constant during all that time. The leading eigenvalue might hover around $\lambda_1 = 4.0$, but its associated vector $v_1$ can change.

It is worth keeping in mind that the traditional SGD with RMSProp does not focus specifically on the directions of negative curvature, so plots like those of Figure 5 and Figure 6 might turn out different if an optimizer focused on exploiting those directions.  We might "exhaust" or "harvest" the most dominant ones, and then find ourselves new directions in which the negative curvature is now the most extreme.

One of the motivations for studying the evolution of the Hessian matrix is that its *stability* is important in certain optimization methods where an estimate of the Hessian is refined over many consecutive steps.  If the actual Hessian changes too drastically, then the estimate is not going to be meaningful, and this may be detrimental to the optimizer.

## C   OPTIMAL STEP SIZES

A strictly-convex loss function $f(\theta)$ has a positive-definite Hessian matrix $H(\theta)$ for all values of $\theta$. That is, all its eigenvalues will be strictly greater than zero.

To perform an update with Newton's method, we update the parameters $\theta_t$ according to

$$\theta_{t+1} = \theta_t - \alpha H(\theta_t)^{-1} g(\theta_t)$$

where $g(\theta_t)$ is the gradient of $f(\theta)$ and $\alpha$ is the learning rate.

In the special case when $f(\theta)$ is quadratic, the Hessian is constant and we can use one Newton update with $\alpha = 1$ to jump directly to the optimum. We can compute what that means in terms of the optimal step size to update $\theta$ along the direction of one of the eigenvector $v_i$.

Let $\{(\lambda_1, v_1), \ldots, (\lambda_d, v_d)\}$ be the eigendecomposition of the Hessian matrix. If we project the gradient in the basis of eigenvectors, we get

$$g(\theta) = \sum_{i=1}^{N} \left[ g(\theta)^T v_i \right] v_i.$$

Note that $H^{-1} v_i = \frac{1}{\lambda_i} v_i$, so we have that

$$H^{-1} g(\theta) = \sum_{i=1}^{N} \left[ g(\theta)^T v_i \right] \frac{1}{\lambda_i} v_i.$$

Thus, when minimizing a strictly-convex quadratic function $f(\theta)$, the optimal step size along the direction of an eigenvector is given by

$$\alpha^* = \arg\min_{\alpha} \mathcal{L} \left( \theta - \alpha \left[ g(\theta)^T v_i \right] v_i \right) = \frac{1}{\lambda_i}. \tag{2}$$

If we are dealing with a strictly-convex function that is not quadratic, then the Hessian is not constant and we will need more than one Newton update to converge to the global minimum. We can still hope that a step size of $1/\lambda_i$ would be a good value to use.

With a deep neural network, we no longer have any guarantees. We can still measure optimal step sizes experimentally, which is what we have done in Section 2.3. We saw in Figure 4 that the optimal step sizes in directions $v_i$ of positive curvature matched rather well with the value of $1/\lambda_i$. It has been suggested in Dauphin et al. (2014) that in directions of negative curvature, the optimal step size could be $1/|\lambda_i|$, but our empirical results are much larger than that. Again, we have to keep in mind that a general theory cannot be extrapolated from only one model and one dataset.

## D   ON ESTIMATING THE HESSIAN

Given that the full Hessian matrix has more than $10^{13}$ coefficients, and that the entire training set has $50000 * 28^2$ coefficients, we might be concerned about whether the value of the Hessian is possible to estimate statistically.

In a way, much like the loss $\mathcal{L}(\theta) = \sum_{n=1}^{N} \mathcal{L}_\theta(x_i, y_i)$ is an exact quantity defined over the whole training set, the Hessian is the same. The notion of an estimator variance would come into play if we estimated $H(\theta)$ from a minibatch instead.

Given the computational costs of evaluating $\mathcal{L}(\theta)$ and $H(\theta)$ on the whole training set every time that the Scipy function `scipy.sparse.linalg.eigsh` wants us to evaluate the Jacobian vector product, we tried to see if it was possible to get away with only using 5% of the training set for that purpose. That 5% has to always contain the same samples, or otherwise we violate assumptions made by Scipy (in a way similar to how the usual quicksort implementation would fail if comparisons were no longer deterministic).

Now $H_{5\%}(\theta)$ is an estimator of $H(\theta)$, and we have verified experimentally that the first elements of the eigenspectrum of those two matrices are close enough for the purposes of our analysis. We did this by comparing LA(10) and SA(10) in both cases, checking the differences between eigenvalues and the angles between the eigenvectors.

## E  SUGGESTION FOR NEW OPTIMIZATION METHOD

Considerable work was required for us to extract negative eigenvalues for every checkpoint of training. This is not a practical thing to do during training, so we want to introduce here the idea of keeping a running approximation of the smallest eigenvector of the Hessian.

We know that the Jacobian vector product $H(\theta)v$ can be evaluated on a minibatch at the same time that we compute the gradient. Some people report an overhead of $4\times$ the computational costs, but we have not measured any benchmarks in that regards.

The smallest eigenvector is a unit vector $v$ that minimizes the value of $m(v) = v^T H(\theta)v$. This is a quadratic in the coefficients of $v$ (along with a constraint on the norm of $v$), and it's something that we can minimize using a method similar to SGD. We can easily see that $\nabla_v m(v) = 2H(\theta)v$, so we can minimize simultaneously $m(v)$ and the usual model loss $\mathcal{L}(\theta)$. This means that we can keep a running estimate $(\tilde{\lambda}, \tilde{v})$ of $(\lambda_d, v_d)$, and we can alternate between one update to $\theta$ with the usual RMSProp/Adam optimizer, and then one update in the direction of $\left[g(\theta)^T \tilde{v}\right] \tilde{v}$. Different learning rates could be used for those updates.

This is not something that we have tried in practice, but it would be the most direct way to implement a training method based on the ideas of this paper.

## F  EXTRA PLOTS

We provide here a few extra plots to accompany Figure 3 and Figure 4 in order to paint a more complete picture.

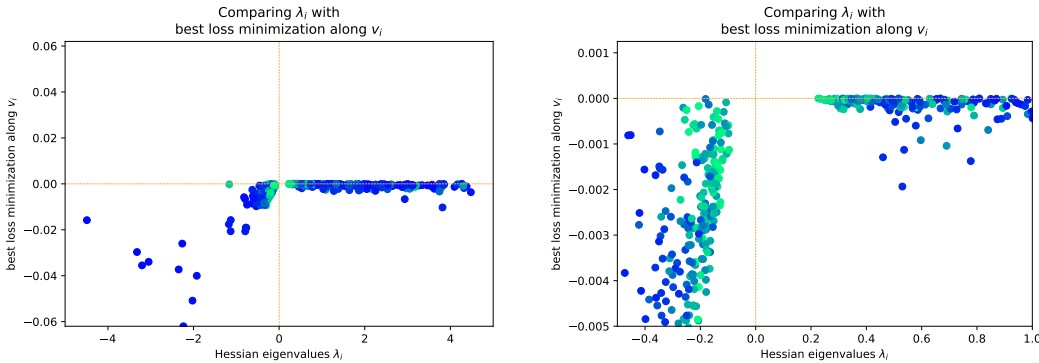

Figure 7: Same as Figure 3 but with a different interval. On the left we have the interval $[-5, 5]$, and on the right we look more closely to the origin. We are using a color gradient to differentiate blue points coming from an earlier training step (closer to $t = 0$) and green points coming later in the optimization. It is not easy to interpret whether there is a significant difference between the two. In terms of Equation (1), these plots show the relation between $\lambda_i$ and $\mathcal{L}(\alpha^*)$.

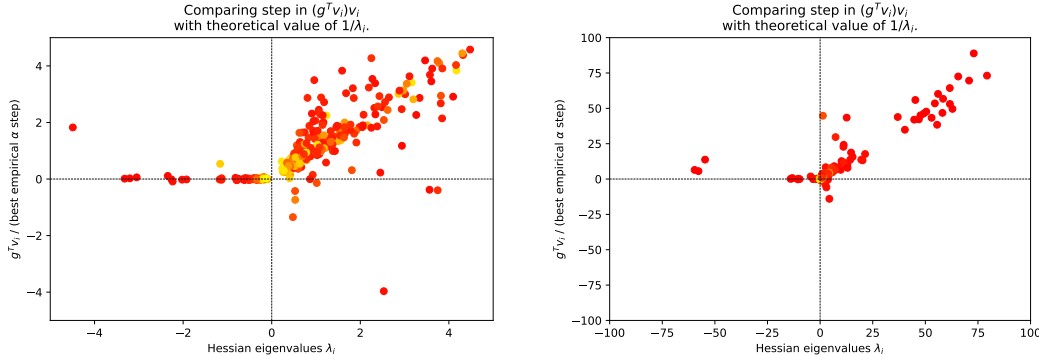

Figure 8: Same as Figure 4 but with a different interval. On the left we have the interval $[-5, 5]$, and on the right we look at $[-100, 100]$. We are using a color gradient to differentiate red points coming from an earlier training step (closer to $t = 0$) and yellow points coming later in the optimization. One of the interesting observations is that the direct correspondence between $1/\alpha^*$ (the $y$-axis) and $\lambda$ (the $x$-axis) seems to hold relatively well for larger values $\lambda$. We knew from Figure 4 that it approximately held for small positive values of $\lambda$, which is something that can also be observed on the left plot here. In terms of the variables involved in Equation (1), these plots show the relation between $\lambda_i$ ($x$-axis) and $1/\alpha^*$ ($y$-axis).

