# OpenReview forum: "Negative eigenvalues of the Hessian in deep neural networks"
_ICLR.cc/2018/Workshop — Accept_

### Official Review · AnonReviewer3 · 2018-03-08
**Interesting suggestions but way to preliminary**

**Rating:** 4
**Confidence:** 3

**Review:**

The main result reported in this contribution is that in a specific convolutional network the loss function decreases more in directions corresponding to negative eigenvalues of the Hessian. The authors suggest that this should be taken into account when designing the step-size used in learning.

It is not clear whether the authors expect their result to hold in learning in other types of neural networks, they set this as a goal for future work. But more importantly it is not clear whether this should hold in a more generic class of non-convex optimization problems  or only for the loss optimization in neural networks.

The authors speculate about ways how to take advantage of their observation and obtained improved algorithms, but do not test them in any setting.

While the questions they are asking are interesting, they do not seem to be sufficiently advanced for this work to be interesting for presentation.

---

### Official Review · AnonReviewer1 · 2018-03-09
**finds existence proof (for 1 model and 1 dataset) where learning rate should be higher than Dauphin et al's suggested 1 / abs(eigenval) for negative curvature directions**

**Rating:** 6
**Confidence:** 4

**Review:**

The main contribution of this work is their simple empirical observation in Figure 4, that the optimal alpha (learning rate) appears to often NOT be 1 / | eigval | when the eigval is negative (contradicting the hypothesis of Dauphin et al. (2014), but instead should be much larger. This seems to this reader to be the only new insight offered by this paper, so not mentioning that connection to that well known related work until Appendix C is concerning (it should be mentioned in the main body).  They mention a way to estimate the negative eigenvector, but it is disappointing that they have “not yet tried this method in practice”.  This paper is clearly premature / under-developed for a standard conference publication; whether suitable for a workshop is debatable.  On the one hand, it would be likely to prompt useful discussions at the workshop.  However, overall, given the lack of trying to use this insight in any actual optimization algorithm on any data, it could easily be seen as more flag planting at this point than a solid contribution.  It could be more suitable if their insights (Figure 4) were established for more than “only one model and one dataset” (as the authors admit).  Also, there has been a fair number of related past work on examining the Hessian (and the tracking of eigenvals over time), but the authors do not cite any.  As one example, LeCun has periodically shown such experiments, including seminal work at NIPS  https://papers.nips.cc/paper/589-automatic-learning-rate-maximization-by-on-line-estimation-of-the-hessians-eigenvectors.pdf and more recent ones on ArXiv.

In short, this might be marginally suitable as a workshop paper, but is overall under-developed and should do a better job of being very clear about its very modest contribution to date (and cite more related work).

---

### Decision · Program_Chairs · 2018-03-20
**ICLR 2018 Workshop Acceptance Decision**

**Decision:**

Accept

**Comment:**

Congratulations, your paper was accepted to the ICLR workshop. While this work hasn't tried a new algorithm yet, the preliminary findings will make a really interesting poster.